# Effect of Niacin on Growth Performance, Intestinal Morphology, Mucosal Immunity and Microbiota Composition in Weaned Piglets

**DOI:** 10.3390/ani11082186

**Published:** 2021-07-23

**Authors:** Shilong Liu, Xiaoping Zhu, Yueqin Qiu, Li Wang, Xiuguo Shang, Kaiguo Gao, Xuefen Yang, Zongyong Jiang

**Affiliations:** 1Institute of Animal Science, Guangdong Academy of Agricultural Sciences, State Key Laboratory of Livestock and Poultry Breeding, Ministry of Agriculture Key Laboratory of Animal Nutrition and Feed Science in South China, Guangdong Provincial Key Laboratory of Animal Breeding and Nutrition, Maoming Branch, Guangdong Laboratory for Lingnan Modern Agriculture, Guangzhou 510640, China; liushilong94@126.com (S.L.); qiuyueqin87@126.com (Y.Q.); wangli1@gdaas.cn (L.W.); gaokaiguo@gdaas.cn (K.G.); jiangzy@gdaas.cn (Z.J.); 2College of Life Science and Engineering, Foshan University, Foshan 528000, China; zhuxiaoping@fosu.edu.cn (X.Z.); xiuguoshang@163.com (X.S.)

**Keywords:** weaned piglets, niacin, growth performance, intestinal morphology, intestinal mucosal immunity, colonic microbiota

## Abstract

**Simple Summary:**

The protective effect of niacin on growth performance and gut health of weaned piglets and the underlying mechanism remains unclear despite it being a common additive in pig diets. The present study aimed to investigate the effect of niacin on growth performance, intestinal morphology, intestinal mucosal immunity, and colonic microbiota in weaned piglets. Our results show that niacin supplementation significantly improved the growth performance in piglets as compared with those given a niacin receptor antagonist. Niacin also significantly improved the relative abundance of beneficial bacteria in the colon and alleviate the inflammatory response in the intestinal mucosa as compared with control piglets and those given a niacin receptor antagonist. These results provide new insight into the beneficial effects of niacin on growth performance and gut health in weaned piglets.

**Abstract:**

This study aimed to investigate the effects of niacin on growth performance, intestinal morphology, intestinal mucosal immunity, and colonic microbiota in weaned piglets. A total of 96 weaned piglets (Duroc × (Landrace × Yorkshire), 21-d old, 6.65 ± 0.02 kg body weight (BW)) were randomly allocated into 3 treatment groups (8 replicate pens per treatment, each pen containing 4 males; *n* = 32/treatment) for 14 d. Piglets were fed a control diet (CON) or the CON diet supplemented with 20.4 mg/kg niacin (NA) or an antagonist for the niacin receptor GPR109A (MPN). The results showed that NA or MPN had no effect on ADG, ADFI, G/F or diarrhea incidence compared with the CON diet. However, compared with piglets in the NA group, piglets in the MPN group had lower ADG (*p* = 0.042) and G/F (*p* = 0.055). In comparison with the control and MPN group, niacin supplementation increased the villus height and the ratio of villus height to crypt depth (*p* < 0.05), while decreasing the crypt depth in the duodenum (*p* < 0.05). Proteomics analysis of cytokines showed that niacin supplementation increased the expression of duodenal transforming growth factor-β (TGF-β), jejunal interleukin-10 (IL-10) and ileal interleukin-6 (IL-6) (*p* < 0.05), and reduced the expression of ileal interleukin-8 (IL-8) (*p* < 0.05) compared with the control diet. Piglets in the MPN group had significantly increased expression of ileal IL-6, and jejunal IL-8 and interleukin-1β (IL-1β) (*p* < 0.05) compared with those in the control group. Piglets in the MPN group had lower jejunal IL-10 level and higher jejunal IL-8 level than those in the NA group (*p* < 0.05). The mRNA abundance of duodenal IL-8 and ileal granulocyte-macrophage colony-stimulating factor (GM-CSF) genes were increased (*p* < 0.05), and that of ileal IL-10 transcript was decreased (*p* < 0.05) in the MPN group compared with both the control and NA groups. Additionally, niacin increased the relative abundance of *Dorea* in the colon as compared with the control and MPN group (*p* < 0.05), while decreasing that of *Peptococcus* compared with the control group (*p* < 0.05) and increasing that of *Lactobacillus* compared with MPN supplementation (*p* < 0.05). Collectively, the results indicated that niacin supplementation efficiently ensured intestinal morphology and attenuated intestinal inflammation of weaned piglets. The protective effects of niacin on gut health may be associated with increased *Lactobacillus* and *Dorea* abundance and butyrate content and decreased abundances of *Peptococcus*.

## 1. Introduction

Weaning piglets from sows is one of the most stressful events in a pig’s life [1]. This process has been shown to be the main cause of intestinal and immune system dysfunction [2], as well as leading to a lower growth rate and severe diarrhea in weaned piglets [3,4,5,6]. Therefore, prevention of intestinal barrier function impairment and inflammation induced by weaning stress may be a potential strategy for the treatment of intestinal injury.

Niacin, also known as vitamin B_3_ or nicotinic acid, is a water-soluble vitamin belonging to the vitamin B family [7,8]. As a nutrient, a study by Real et al. showed that diets supplemented with 50 g/ton niacin significantly improved ADG and ADFI in nursery pigs [9]. At the pharmacological dose level (0.5–3 g/day), niacin alleviates inflammation in human monocytes by decreasing the secretion of tumor necrosis factor-α (TNF-α) and interleukin-6 (IL-6) through a G protein-coupled receptor 109A (GPR109A)-dependent pathway [10,11]. Additionally, niacin has a beneficial effect by attenuating the inflammatory mechanisms involved in atherosclerosis pathology and plaque stabilization [12]. Kwon et al. reported that niacin attenuates lung inflammation and improves survival during sepsis in rats by downregulating the nuclear factor activated B cell κ-light chain enhancement (NF-κB) signaling pathway [13]. Furthermore, literature studies indicate that niacin can maintain gut health. Feng et al. suggested that niacin increases intestinal immune function in fish, partly by downregulating the expression of TNF-α, interleukin-1β (IL-1β), interferon-γ (IFN-γ), and interleukin-8 (IL-8) and upregulating the expression of interleukin-10 (IL-10) and transforming growth factor-β (TGF-β) compared niacin deficiency group [14]. Salem et al. demonstrated that niacin decreased colonic myeloperoxidase (MPO) activity and TNF-α expression induced by iodoacetamide in a GPR109A-dependent manner [15]. However, studies regarding the protective effects of niacin on the intestinal health in weaned piglets are limited.

G protein-coupled receptor 109A (GPR109A) is a receptor for niacin, butyrate, and hydroxybutyric acid [16], and is expressed on the surface of adipocytes [17], macrophages [18], neutrophils [19], and intestinal epithelial cells [20]. Recent studies have shown that GPR109A agonists can exert beneficial effects on the gut, including promotion of gut epithelial integrity, suppression of intestinal inflammation, and regulation of colonic bacteria [21,22]. As an essential receptor for niacin, GPR109A signaling protects against dextran sulphate sodium (DSS)-induced colitis; however, niacin fails to maintain gut health in the presence of a GPR109A antagonist [15]. Singh et al. found that the binding of niacin to GPR109A promotes anti-inflammatory properties in colonic macrophages and dendritic cells, induced differentiation of Treg and IL-10-producing T cells, and increased the expression of colonic IL-18, which contributed to protecting intestinal barrier function [21]. Moreover, Chen et al. demonstrated that GPR109A plays a role in regulating the structure of the gut microbiota and the microbial population [23]. Further, Bhatt et al. showed that GPR109A signaling inhibits the microbiota-induced production of inflammatory cytokines (IL-23) in the colon, suppressing IL-23-mediated intestinal inflammation [24]. Nevertheless, limited information regarding whether niacin directly regulates intestinal health in weaned piglets via GPR109A.

Therefore, this study aimed to investigate the effects of niacin supplementation or niacin receptor antagonist administration on the growth performance, intestinal morphology, intestinal immunity, and colonic microbiota composition in weaned piglets.

## 2. Materials and Methods

All animal protocols in the present study were performed in accordance with the Guidelines for the Care and Use of Animals for Research and Teaching following approval by the Animal Care and Use Committee of Guangdong Academy of Agricultural Science (authorization number GAASIAS-2016-017).

### 2.1. Animals and Experimental Treatments

A total of 96 weaned piglets (Duroc × (Landrace × Yorkshire), 21 d old, barrow) with an initial weaning weight of 6.65 ± 0.02 kg were randomly allocated into 3 dietary treatments. Each treatment consisted of 8 replicate pens, with 4 piglets per pen (*n* = 32 piglets per treatment). The piglets fed a basal diet were considered as the control group (CON), and the other groups were fed the basal diet supplemented with 20.4 mg niacin/kg diet (NA), or mepenzolate bromide, an antagonist for the niacin receptor GPR109A (MPN) for a 14-d feeding trial. Piglets in the MPN group were administered orally with 10 mg/kg/day niacin receptor antagonist every day in the morning, the dose of the antagonist was based on previous studies [25,26], and piglets in CON and NA groups received the same amount of water. The basal diet was formulated to meet the nutrient recommendations of the National Research Council 2012 (NRC 2012), and the diet compositions and nutrient profiles are presented in Table 1.

All pigs were allotted in cages containing 4 piglets with slatted floors at the experimental farm of Institute of Animal Science, Guangdong Academy of Agricultural Sciences, China. The piglets were fed four times a day (at 8:00 a.m., 12:00 p.m., 16:00 p.m. and 20:00 p.m.) with the prepared diet in feeding troughs and had ad libitum access to feed and water. The amount of feed consumed per pen was recorded every day to determine the average daily feed intake (ADFI) on an 88.53% dry matter basis. All of the piglets were weighed individually at the beginning and the end of the experiment to calculate the average daily gain (ADG). The gain to feed ratio (G/F) was calculated based on the weight gain and feed intake (based on 88.53% dry matter) during the whole experiment period. At the end of the study (at 8:00 a.m. of the 15th day), piglets with body weight (BW) closest to average BW of each pen was selected (totally 24 pens) and transferred to the slaughterhouse within the Institute of Animal Science, Guangdong Academy of Agricultural Sciences, China. Blood samples were collected from the anterior vena cava of the selected piglets using ethylenediaminetetraacetic acid (EDTA)-anticoagulant tubes. Then piglets were anesthetized with sodium pentobarbital (40 mg/kg BW) and sacrificed. The whole intestine was quickly removed and placed on a cold tray to collect the duodenum, jejunum and ileum. Intestinal mucosa samples were gathered by scraping with sterile glass microscope slides, and intestinal digesta (including jejunum, ileum and colon) were collected and quickly frozen in liquid nitrogen and stored at −80 °C for analysis. The samples of serum were gathered by centrifugation at 3500× *g* for 15 min at 4 °C and then stored at −80 °C for further analysis.

### 2.2. Feed Analysis

In the present study, dry matter (GB/T6435—2014), crude protein (GB/T6432—2014), crude fat (GB /T6433—2014), crude fiber (GB /T6434—2014), crude ash (GB /T6438—2014) in the feed were determined according to the mentioned procedures of National Standard Methods (China).

### 2.3. Intestinal Morphology

Sections of approximately 2.0 cm in length of the middle duodenum, middle jejunum and distal ileum were rinsed with ice-cold PBS and fixed in 4% paraformaldehyde for morphometric evaluation and histochemical staining. The fixed samples were embedded in paraffin, and 4-μm cross sections from each specimen were mounted on slides coated with polylysine, deparaffinized, rehydrated, and then stained with hematoxylin-eosin (HE) for intestinal morphological examination. HE-stained slices were scanned by a digital brightfield microscope scanner (Pannoramic 250, 3D HISTECH, Budapest, Hungary). Twenty well-oriented and intact villi and adjacent crypts were randomly selected to determine the villus height and crypt depth of each segment using slide viewer software (Case Viewer 2.3, 3D HISTECH, Budapest, Hungary), and the villus height-to-crypt depth ratio (VCR) was calculated.

### 2.4. Protein Chip

QAP-CYT-1 kits (Ray Biotech, Guangzhou, China) were used for cytokine detection and quantitation. Mucosal and serum samples were treated with 1× lysis buffer containing protease inhibitor cocktail, and the protein concentration of the lysate was determined using a BCA assay. For the detection of cytokines, samples were incubated on glass slides overnight at 4 °C, washed using a Thermo Scientific Well Wash Versa (Waltham, MA, USA), and incubation with Cy3 Equivalent Dye-streptavidin. After removal of unbound material by washing, an Inno Scan 300 (Innopsys, Parcd’ Activités Activestre, Carbonne, France) was used for fluorescence detection (scanning signals and Cy3 or green channels; excitation frequency = 532 nm).

### 2.5. Quantitative Real-Time PCR

Total RNA from intestinal mucosa (including duodenum, jejunum and ileum) samples was extracted with Trizol reagent (Invitrogen, Carlsbad, CA, USA). RNA purity and yield were evaluated using a Nano Drop 1000 (Thermo Fisher Scientific, Waltham, MA, USA) and RNA integrity was determined by electrophoresis on 1% agarose gels. A total of 1 μL of total RNA was used to synthesize cDNA using a PrimeScript™II 1st Strand cDNA Synthesis Kit (Takara, Dalian, China). SYBR green I (Bio-Rad, Hercules, CA, USA), 10-fold diluted cDNA and gene-specific primers (Table 2) in a final volume of 20 μL were used to perform qPCR analyses in triplicate. The qPCR conditions were 95 °C × 3 min followed by 40 cycles of amplification (95 °C × 15 s, 60 °C × 30 s, and 72 °C × 30 s). In the present study, to normalize expression data, β-actin and glyceraldehyde-3-phosphate dehydrogenase (GAPDH) were used as the internal control genes according to a previous study showing that the expression stability was evaluated by the M value and pairwise variations of GENorm (Version 3.5; Primer Design Ltd., Southampton, Hampshire, UK) [27]. The results showed that β-actin had a lower M value than GAPDH, both below 1.5. Thus, β-actin was ranked as the most stably expressed gene. Additionally, the pairwise variations of β-actin from intestinal mucosa samples were below the threshold (0.15) that required the inclusion of an additional normalization gene. Thus, β-actin was used for normalization in the present study. The fold changes in target gene expression of piglets in treatment groups were normalized to β-actin and relative to the expression of those in CON group; fold changes were calculated for each sample using the 2^−ΔΔCt^ method, where ΔΔCT = (CT, Target − CT, β-actin) Treatment − (Average CT, Target − Average CT, β-actin) Control [28]. The relative mRNA expression levels of the target genes were determined using the 2^–^^ΔΔCt^ method, and data for each target transcript were normalized to that of the control piglets.

### 2.6. Gut Microbiota

Total genomic DNA was extracted from colonic digesta samples using the QIAamp DNA kit (Qiagen, Dusseldorf, Germany) follow the manufacturer’s instructions and DNA quality was determined by electrophoresis on 1% agarose gels. The V3-V4 region of 16S rRNA was amplified by PCR using specific barcode primers of all colon content samples. A total volume of 30 µL was used in PCR reactions and the amplification products were purified. Ion Plus Fragment Library Kit 48 rxns (Thermo Scientific, Waltham, MA, USA) was used to construct the library and Qubit@ 2.0 Fluorometer (Thermo Scientific, Waltham, MA, USA) was used to detect whether the library was qualified. After that, Cutadapt (V1.9.1, http://cutadapt.readthedocs.io/en/stable/, accessed on 18 December 2020) was used to cut the low-quality part of the reads, and then the sample data were separated from the reads obtained using a barcode, the raw data were obtained (raw reads) after the above processing steps. The reads obtained after the above processing needed to be processed. Processing of removing of the chimera sequence was done and the read sequence was compared with the species annotation database to detect the chimera sequence; finally, the chimera sequence is removed, and the final valid data (Clean Reads) were obtained. All clean reads of all samples were clustered using software Uparse (v7.0.1001 http://www.drive5.com/uparse/, accessed on 18 December 2020) to become OTUs; Qiime software (version 1.9.1., University of California, San Diego, CA, USA) was used to calculate Chao1, Shannon, ACE, and principal coordinates analysis (PCoA). The function prediction of cecal microbiota was conducted using Phylogenetic Investigation of Communities by Reconstruction of Unobserved States (PICRUST) [29].

### 2.7. Gas Chromatographic Analysis

The contents of short-chain fatty acids (SCFAs) (including acetic acid, propionic acid, butyric acid, isobutyric acid, valeric acid, isovaleric acid) in colon digesta of weaned piglets were determined using gas chromatography-mass spectrometry. Briefly, 0.1 g of colonic digesta sample was mixed with 1 mL of 5 mmol NaOH and 50 mL of 0.5 μg/mL^−1^ hexanoic acid 3. The supernatant was collected after centrifuging at 12,000× *g*, 4 °C for 15 min. Then, the mixture (including the above-mentioned supernatant, 250 μL of propanol/pyridine (*V*/*V*, 3:2), 150 μL of water, 50 μL of propyl chloroformate) was mixed into derivatives for 5 min and extracted twice with n-hexane. A total of 10 mg anhydrous Na_2_SO_4_ was used for dehydration, and 150 mg supernatant was collected for analysis.

### 2.8. Performance Liquid Chromatography—Mass Spectrometry

The collected 200-mg samples (include jejunum, ileum, colon contents) were mixed with 750 μL methanol/acetonitrile/water (*v*/*v*/*v*, 2/2/1) solution, and 10 μL of internal standard solution (homogenate). The separation of the samples used a Waters ACQUITY UPLCI-Class system. The chromatographic column (Thermo Fisher Scientific, Waltham, MA, USA) used water, HSS T3 (2.5 μm, 2.1 mm × 100 mm); column temperature: 45 °C; mobile phase (liquid A, 0.3% formic acid aqueous solution; liquid B, methanol). After that, a 5500 QTRAP mass spectrometer (AB SCIEX, Framingham, MA, USA) was used to conduct mass spectrometric ionization mode in positive ion mode: ESI+, ion spray voltage: +4500 V, ion source temperature: 550, air curtain gas (CUR) 40 psi, atomized gas (GS1) 55 psi, auxiliary gas (GS2) 55 psi, use mode to detection pairs and conditions included: parent ion, 124.0; daughter ion, 80.2; dwell time, 15.0; decluster potential, 109.0; collision energy, 30. The standard curve solution was prepared by eddy mixing 150 μL of the standard working solution, 600 μL of methanol/acetonitrile (volume ratio 1:1) and 10 μL of the internal standard solution in a 1.5 mL centrifuge tube. The standard curve was established using the isotope internal standard method.

### 2.9. Statistical Analysis

Data were analyzed using SPSS 20.0 (SPSS Inc., Chicago, IL, USA) and are presented as mean ± SD. The results were assessed by one-way analysis of variance (ANOVA) followed by Tukey’s test. Differences were considered significant at *p* < 0.05.

## 3. Results

### 3.1. Effect of Niacin on Growth Performance of Weaned Piglets

As shown in Table 3, piglets fed NA or MPN showed no difference in ADG, ADFI, G/F or diarrhea incidence compared with those fed the control diet. However, piglets in the MPN group had lower ADG (*p* = 0.042) and G/F (*p* = 0.055) compared with the control group.

### 3.2. Niacin Content

As shown in Table 4, compared with piglets in the CON and MPN groups, the piglets in NA group showed significantly increased niacin content in the jejunum digesta and nicotinamide content in serum

### 3.3. Effect of Niacin on Intestinal Morphology of Weaned Piglets

Compared with the piglets in the control and MPN groups, piglets fed diet supplemented with niacin showed significantly increased villus height and villus height-to-crypt depth ratio (VCR), and reduced crypt depth (*p* < 0.05) in the duodenum. Moreover, in comparison with the control diet, MPN significantly decreased jejunal VCR and ileal crypt depth, and increased jejunal crypt depth (*p* < 0.05) (Figure 1).

### 3.4. Effect of Niacin on Intestinal Immunity of Weaned Piglets

Proteomics analysis of cytokines shown in Figure 2A–C indicated that niacin supplementation significantly improved the expression of TGF-β in duodenal mucosa (*p* < 0.05), IL-10 in jejunal mucosa (*p* < 0.05), IL-6 in ileal mucosa (*p* < 0.05), but reduced the expression of IL-8 in ileal mucosa (*p* < 0.05) when compared with the control diet. Piglets in the MPN group had significantly increased expression of IL-6 in ileal mucosa (*p* < 0.05), IL-8 and IL-1β in jejunal mucosa (*p* < 0.05) when compared with those in the control group, the piglets in MPN group showed significantly reduced expression of IL-10 (*p* < 0.05), and increased IL-8 (*p* < 0.05) in jejunal mucosa compared with those in the control group. Additionally, the relative protein abundance of cytokines had no significant difference in serum among the three groups (Figure 2D).

The mRNA abundance of duodenal IL-8 (*p* < 0.05) and ileal GM-CSF genes (*p* < 0.05) were increased, and the abundance of ileal IL-10 transcript (*p* < 0.05) was decreased in the MPN group when compared with the control and NA groups (Figure 3A,C). No significant difference in the mRNA abundance of cytokines in jejunal mucosa was found among the treatment groups (Figure 3B).

### 3.5. Effect of Niacin on Colonic Microbiota and Short-Chain Fatty Acids of Weaned Piglets

Results of 16s RNA sequencing of colonic microbes show that there were 1691 OTUs among the three groups, with 840 overlapping. The MPN group had the largest number of OTUs (Figure 4A). PCA analysis revealed significant differences among all groups (Figure 4B). Shannon, Chao 1, and ACE index analysis indicated that there was no significant difference in colonic microbiota richness or diversity among the groups (Figure 4C–E).

The compositions of the 10 most abundant phyla and genera in the colonic digesta were shown in Figure 5A,B. Bacteria that were differentially enriched were shown in Figure 6. Compared with piglets in CON and MPN, piglets in NA showed significantly increased relative abundance of *Dorea* (genus). Compared with piglets in CON, piglets in NA showed significantly decreased relative abundance of *Peptococcus* (genus). Moreover, compared with the piglets in MPN group, piglets in NA group showed significantly increased relative abundance of *Lactobacillus* (genus).

As shown in Figure 7, compared with piglets fed the CON diet, piglets fed diet supplemented with niacin or niacin receptor antagonist had no significant difference in colonic SCFAs levels. However, piglets fed diet supplemented with niacin showed significant increases in the levels of colonic butyrate, valerate and iso valerate compared with the GPR109A antagonist group (*p* < 0.05). Additionally, the results of microbial function predicted by PICRUST show that animal symbionts were significantly enriched in NA group as compared with the control and MPN groups (Figure 8).

## 4. Discussion

Weaning stress is associated with poor gut health, such as intestinal inflammation and unbalanced microbiota composition [31,32,33]. Several recent studies have shown that niacin effectively relieves the inflammatory response and maintains good gastrointestinal tract health. In the present study, niacin ensured growth performance and intestinal morphology and regulated mucosal immunity and colonic microbiota of weaned piglets; however, the beneficial effects of niacin are suppressed after a GPR109A antagonist administration.

The results in the present study showed no significant improvement in the growth performance of weaned piglets in the niacin supplementation group compared with that in the control group. Consistently, previous studies have shown that dietary niacin has no effect on ADG or ADFI in early-weaned and growing pigs [34,35]. In contrast, Real et al. indicated that supplementation with 50 g/ton niacin in the diet significantly improves the growth performance of pigs [9]. The most likely explanations for these discrepancies in the effect of niacin in weaned piglets may be due to the differences in niacin content and the experimental period in previous studies. Additionally, blocking GPR109A using mepenzolate bromide significantly reduced ADG and G/F in weaned piglets in comparison with the niacin group. Similarly, a previous study showing that a significant decline in body weight in rats following administration of a GPR109A antagonist was observed [15]. Collectively, these results might suggest that niacin bind to GPR109A ensured the growth performance of weaned piglets.

It is well known that the small intestine is a crucial digestive and absorptive organ for multifarious dietary nutrients [36]. Intestinal capabilities and morphology are evidenced to be extremely important in piglet growth [37,38], and these influences could extend into adulthood [39]. Large numbers of crypt–villus units covering the epithelium of the small intestine are considered the main representation of intestinal morphology. A larger villus height indicates a better nutrient absorption capacity [40,41]. Herein, our results showed that a smaller villus height, larger crypt depth, and smaller VCR (the ratio of villus height to crypt depth) values were observed in the jejunum and duodenum of piglets in the GPR109A antagonist group than those in the niacin supplementation group. Consistently, a recent study reported that GPR109A^−/−^ mice have severe intestinal structural damage [42]. Our results thus indicated that blocking GPR109A had a negative effect on intestinal morphology in weaned piglets.

The intestinal mucosa is considered the first line of defense to prevent the entry of pathogens into the intestinal tract, where high incidence of inflammatory response reacted. The inflammatory response is important for the systemic immune response, which is primarily mediated by cytokines. Increasing evidence has demonstrated the anti-inflammatory effect of niacin in vivo experimental models. Feng et al. suggested that niacin promotes intestinal immune function partly by regulating cytokine expression in fish, including TNF-α, IL-1β, IFN-γ, IL-8, IL-10, and TGF-β which may be involved in the NF-kB pathway in fish [14]. Kwon et al. demonstrated that niacin reduces the expression of IL-6 and TNF-α in serum, thereby reducing lung inflammation [13]. Similarly, our results show that piglets in the niacin supplementation group had significantly reduced expression of IL-8 and IL-6 and increased expression of TGF-β in small intestinal mucosa as compared with those in the control group. Additionally, it has been reported that the beneficial effects of niacin are mediated through GPR109A. Salem et al. showed that that the binding of niacin to GPR109A promoted anti-inflammatory properties in colonic macrophages and dendritic cells [15]. Furthermore, Singh et al. found that the binding of niacin to GPR109A promotes anti-inflammatory properties in colonic macrophages and dendritic cells [21]. In accordance, our data show that blockade of GPR109A significantly increased the expression levels of IL-8, tended to increase the expression levels of IL-6, GM-CSF, and TNF-α, and decreased the expression levels of IL-10 in small intestinal mucosa as compared with those in the niacin supplementation group. Moreover, the relative mRNA abundances of IL-8 and GM-CSF were significantly increased, and that of IL-10 was significantly decreased, in the GPR109A antagonist group as compared with those in the niacin supplementation group. Thus, these results suggested that niacin supplementation attenuated the intestinal inflammation through GPR109A signaling.

Previous studies have shown that niacin can improve microbial activity in the rumen and promote the synthesis of microbial protein, thereby changing the fermentation pathway of carbohydrates and increasing the concentration of total volatile fatty acids in the rumen [43,44,45]. Furthermore, a study by Feng et al. showed that niacin improved the microbiota and short-chain fatty acid concentrations in the colon of piglets [46]. Similarly, in the present study, dietary supplementation with niacin dramatically decreased the abundances of *Peptococcus* at the genus level compared with those in the control group. Previous studies have indicated that *Peptococcus* are harmful bacteria in high numbers [47], and are particularly pathogenic in bone and joints in humans [48]. Moreover, the abundance of *Peptococcus* is significantly enriched in patients with ulcerative colitis [49]. Additionally, in the present study, the piglets in the niacin supplementation group had a remarkably increased abundance of *Lactobacillus* and *Dorea* at the genus level as compared with those in piglets in the GPR109A antagonist group. *Lactobacillus*, which is the main short-chain fatty acids (SCFAs), especially butyrate-producing bacteria in the colon, has been repeatedly reported to be crucial for the prevention of pathogen infection and alleviation of intestinal inflammation [50,51]. *Dorea* has been associated both detrimentally and advantageously with inflammatory diseases [52,53,54]. Consistent with our findings, a previous study showed that the relative abundance of harmful bacteria is upregulated in GPR109A^−/−^ rats [23]. Collectively, these finding suggested that niacin supplementation alleviated intestinal inflammation via an increase in the abundance of *Lactobacillus* and *Dorea* and a reduction in the relative abundance of *Peptococcus*. To our best knowledge, this study is the first to investigate the effects of dietary niacin supplementation and GPR109A antagonist administration on gut microbiota in weaned piglets.

Furthermore, the results in the present study indicated that supplementation with niacin significantly increased butyrate, valerate and iso valerate contents in colon digesta compared with the GPR109A antagonist group, which was supported by the increased abundance of *Lactobacillus*. SCFAs, which are produced by gut microbiota fermentation of nondigestible dietary fiber, can promote the absorption of water and electrolyte in colon, maintain the integrity of intestinal barrier and relieve diarrhea [55,56,57]. Additionally, butyrate is an efficient agonist for GPR109A [21,58]. Thus, based on these results, we suggested that niacin-mediated alleviation of intestinal inflammation of weaned piglets may be linked to changes in the gut microbiota and the increased butyrate content in the colon.

## 5. Conclusions

In conclusion, the results of the present study indicated that niacin may efficiently ensure intestinal morphology and attenuate intestinal inflammation of weaned piglets. The protective effects of niacin on gut health may be associated with the increased *Lactobacillus* and *Dorea* abundance and butyrate content and the decreased abundances of *Peptococcus*.

## Figures and Tables

**Figure 1 animals-11-02186-f001:**
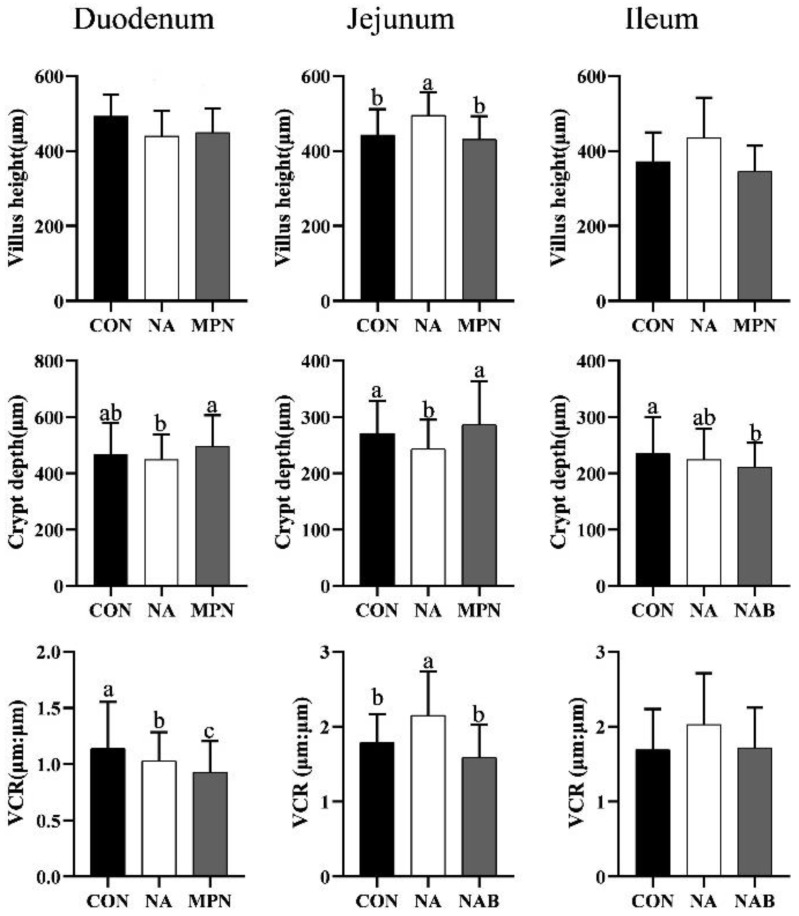
Effect of niacin on intestinal morphology in weaned piglets. Values are expressed as the mean ± SD, *n* = 8/treatment. ^a, b, c^ Values with different superscripts in the same row differ significantly (*p* < 0.05). VCR, villus height-to-crypt depth ratio; CON, control group; NA, niacin group; MPN, GPR109A antagonist group.

**Figure 2 animals-11-02186-f002:**
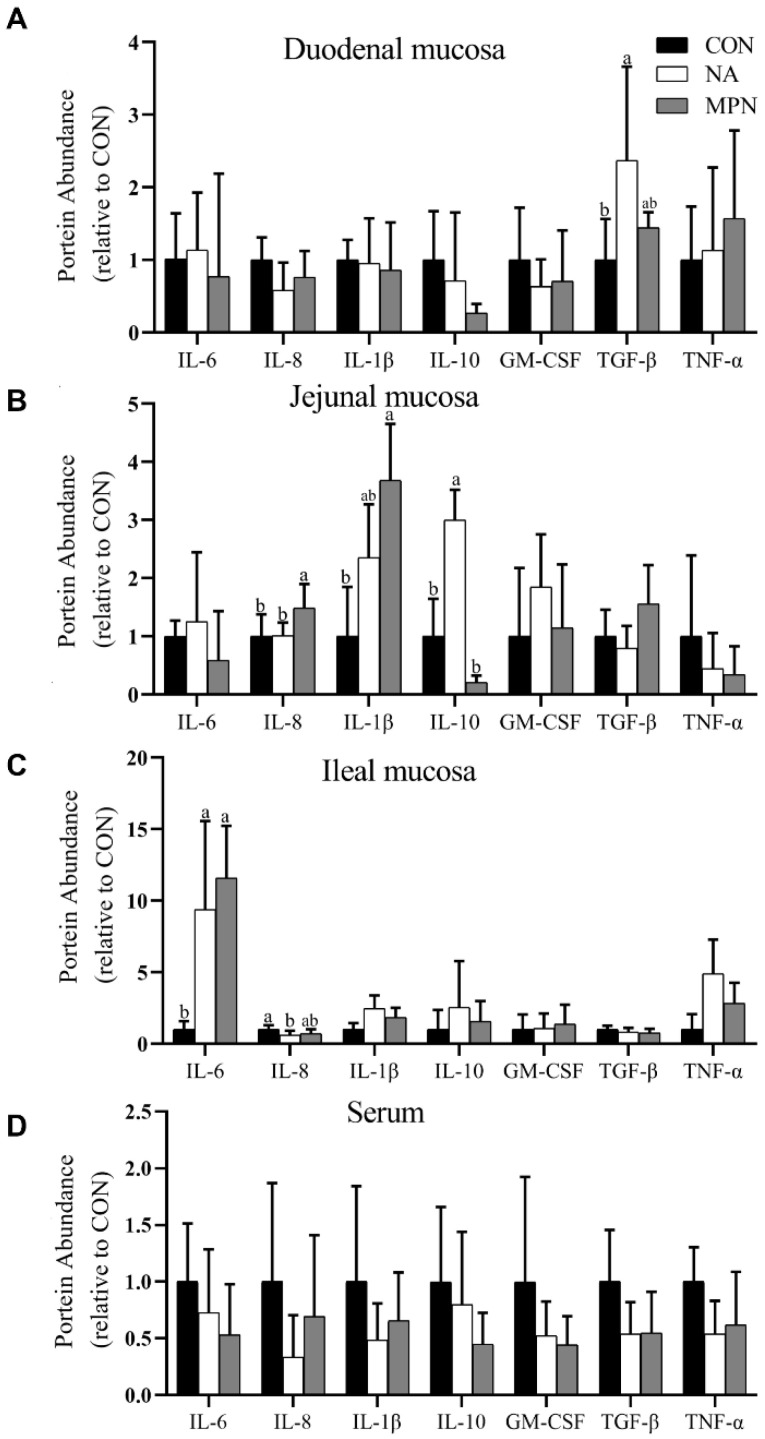
Effect of niacin on the expression of proteins related to the inflammatory response in the small intestinal mucosa and serum of weaned piglets; (**A**) Duodenal mucosa; (**B**) Jejunal mucosa; (**C**) Ileal mucosa; (**D**) Serum. Values are expressed as the mean ± SD, *n* = 8/treatment. ^a, b^ Values of different superscripts in the same row mean significantly different (*p* < 0.05). IL-6, interleukin-6; IL-8, interleukin-8; IL-1β, interleukin-1β; IL-10, interleukin-10; GM-CSF, granulocyte-macrophage colony stimulating factor; TGF-β, transforming growth factor-β; TNF-α, tumor necrosis factor-α; CON, control group; NA, niacin group; MPN, GPR109A antagonist group.

**Figure 3 animals-11-02186-f003:**
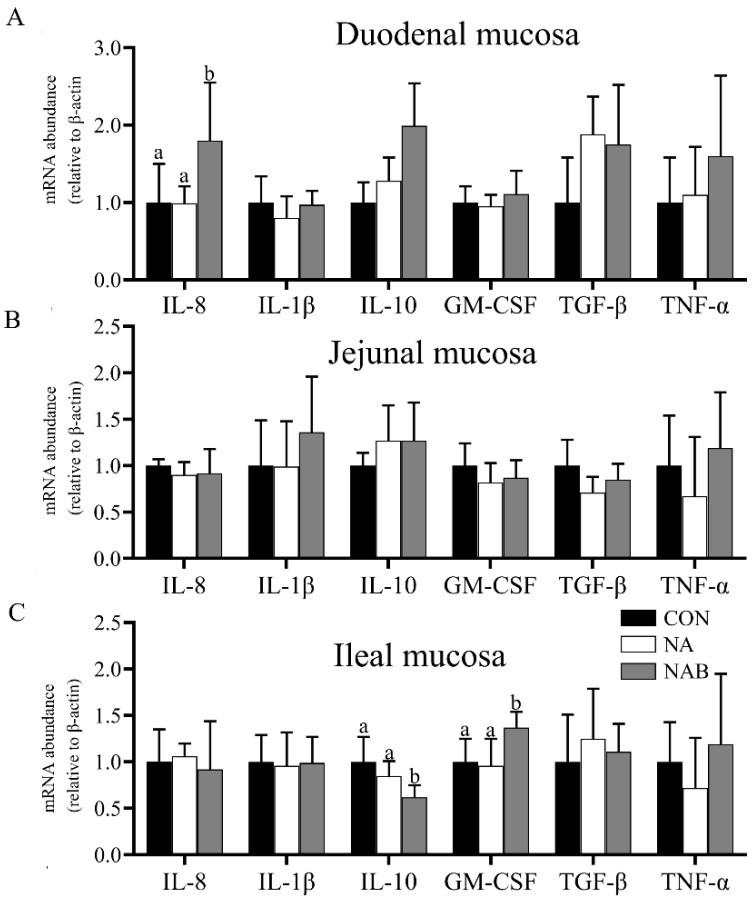
Effect of niacin on the mRNA expression genes related to the inflammatory response in the small intestinal mucosa of weaned piglets; (**A**) Duodenal mucosa; (**B**) Jejunal mucosa; (**C**) Ileal mucosa. Values are expressed as the mean ± SD, *n* = 8/treatment. IL-8, interleukin-8; IL-1β, interleukin-1β; IL-10, interleukin-10; GM-CSF, granulocyte-macrophage colony stimulating factor; TGF-β, transforming growth factor-β; TNF-α, tumor necrosis factor-α. ^a, b^ Values with different superscripts in the same row differ significantly (*p* < 0.05). CON, control group; NA, niacin group; MPN, GPR109A antagonist group.

**Figure 4 animals-11-02186-f004:**
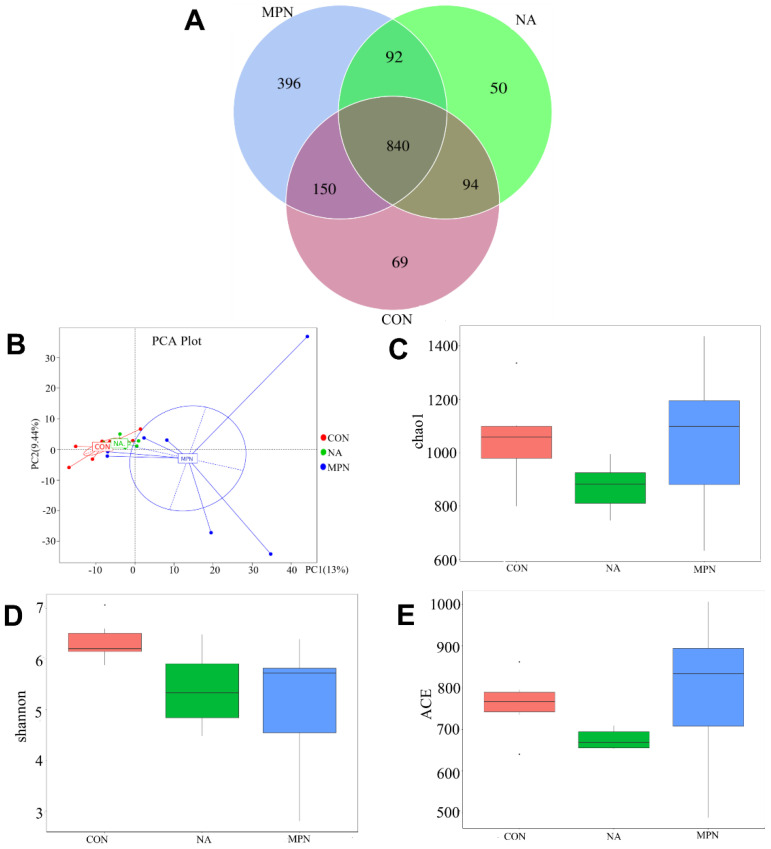
Effect of niacin on colonic microbial communities in weaned piglets. Venn diagram of OTU (**A**), PCA (**B**), Chao 1 index (**C**), Shannon index (**D**), and ACE index (**E**) showing the α- and β-diversity of colonic microbiota. CON, control group; NA, niacin group; MPN, GPR109A antagonist group.

**Figure 5 animals-11-02186-f005:**
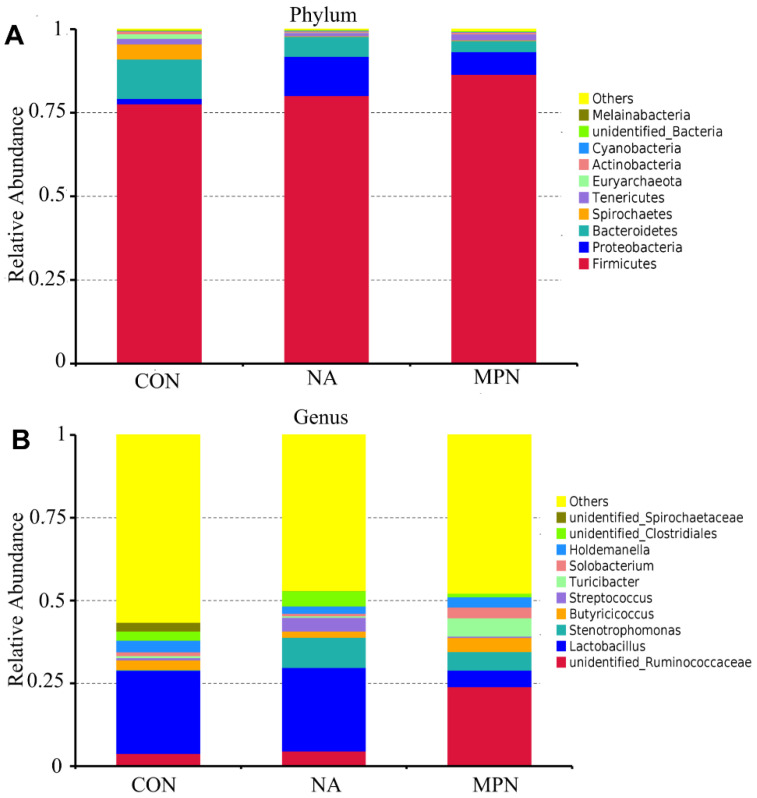
Effect of niacin on colonic microbial communities in weaned piglets. The relative composition of the most abundant 10 species in the colonic contents at the phylum (**A**) and genus (**B**) levels in the different groups. CON, control group; NA, niacin group; MPN, GPR109A antagonist group.

**Figure 6 animals-11-02186-f006:**
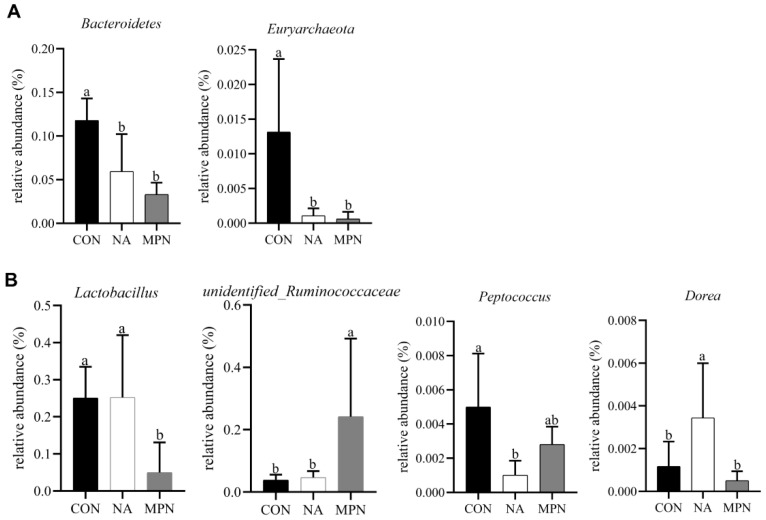
Effects of niacin on the relative abundance of intestinal bacteria in weaned piglets. ^a, b^ Values of different superscripts in the same row mean significantly different (*p* < 0.05). Values are expressed as the mean ± SD, *n* = 8/treatment. Comparison of differential colonic microbiota at the phylum (**A**) and genus (**B**) levels. CON, control group; NA, niacin group; MPN, GPR109A antagonist group.

**Figure 7 animals-11-02186-f007:**
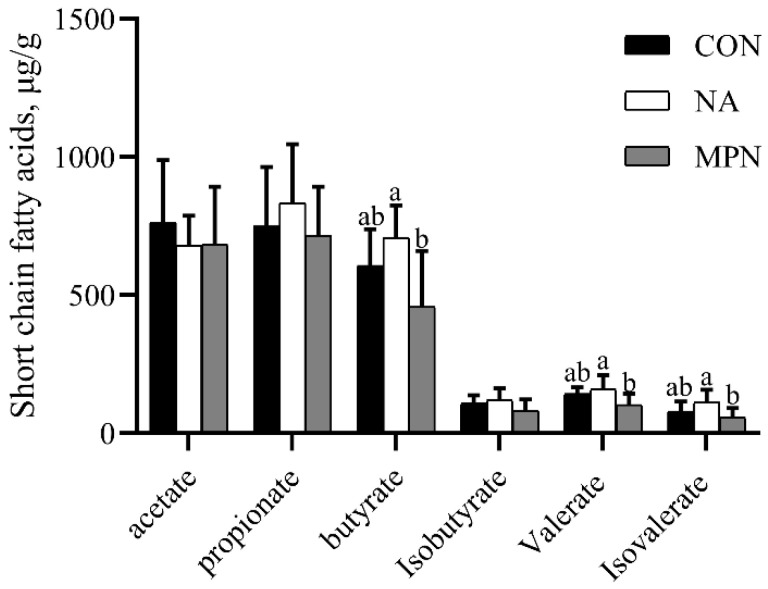
Effects of niacin on SCFAs levels in the colonic contents of weaned piglets. ^a, b^ Values of different superscripts in the same row mean significantly different (*p* < 0.05). Values are expressed as the mean ± SD, *n* = 8/treatment. CON, control group; NA, niacin group; MPN, GPR109A antagonist group.

**Figure 8 animals-11-02186-f008:**
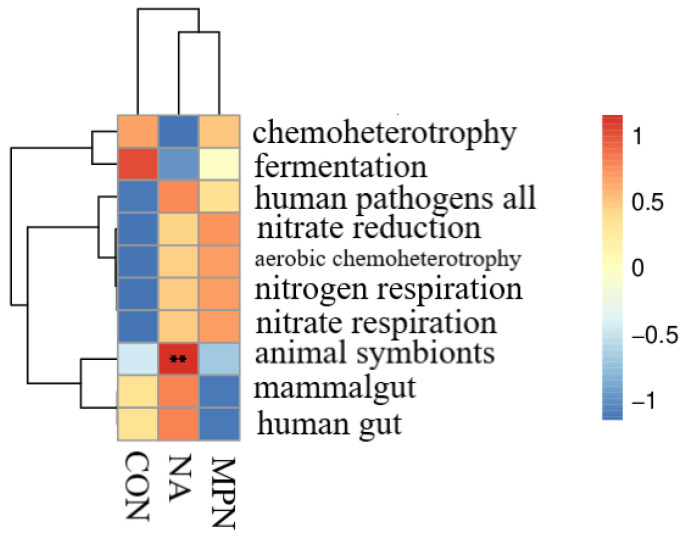
Prediction of microbial function of colonic microbiota in weaned piglets by PICRUST. Heatmap of the predicted microbial function in the three treatment groups. CON, control group; NA, niacin group; MPN, GPR109A antagonist group. ** *p* < 0.01.

**Table 1 animals-11-02186-t001:** Formulation and chemical composition of the basal diet (on an 88.53% dry matter basis).

Item	Diets ^1^	
	CON/MPN	NA
Ingredient, %		
Corn	34.00	34.00
Expanded corn	18.00	18.00
Soybean meal	9.50	9.50
Expanded soybean	15.00	15.00
Lactose	2.00	2.00
Whey powder	8.00	8.00
Fish meal	5.00	5.00
Soybean oil	2.00	2.00
Soybean hull	0.31	0.31
Limestone	0.60	0.598
Lysine 98.5%	0.80	0.80
Methionine	0.30	0.30
L-Threonine	0.30	0.30
Valine	0.13	0.13
L-Tryptophan	0.10	0.10
Monocalcium phosphate	1.50	1.50
Salt	0.25	0.25
60% choline chloride	0.15	0.15
Acidifier	0.05	0.05
Phytase	0.01	0.01
Niacin	0	0.002
Premix ^2^	2.00	2.00
Total	100.00	100.00
Estimated energy and nutrient composition^-^		
GE, kcal/kg	3526.00	3526.00
ME, kcal/kg	3395.01	3395.01
NE, kcal/kg	2610.93	2610.93
Crude protein, %	19.00	19.00
Crude fat, %	8.00	8.00
SID ^3^ lys, %	1.49	1.49
SID thr, %	0.86	0.86
SID try, %	0.27	0.27
SID met + cys, %	0.83	0.83
Calcium, %	0.88	0.88
Total phosphorus, %	0.68	0.68
STTD ^4^ phosphorus, %	0.47	0.47
Analyzed energy and nutrient composition		
GE, kcal/kg	3524.72	3524.72
Crude protein, %	19.30	19.30
Crude fat, %	8.74	8.74
Crude fiber, %	3.39	3.39
Crude ash, %	1.48	1.48
The content of niacin in diets, mg/kg	9.92	30.32

^1^ CON = Control group; NA = niacin group; MPN = GPR109A antagonist group. Piglets in the MPN group were administered orally with 10 mg/kg/day niacin receptor antagonist every day in the morning, and piglets in CON and NA groups received the same amount of water. ^2^ Provided vitamin and mineral premix per kg diet: vitamin A, 2400 IU; vitamin D_3_, 2800 IU; vitamin E, 200 IU; vitamin K_3_, 5 mg; vitamin B_12_, 40 μg; vitamin B_1_, 3 mg; vitamin B_2_, 10 mg; pantothenic acid, 15 mg; folic acid, 1 mg; vitamin B_6_, 8 mg; biotin, 0.08 mg; Fe (FeSO_4_·H_2_O), 120 mg; Cu (CuSO_4_·5H_2_O), 16 mg; Mn (MnSO_4_·H_2_O), 70 mg; Zn (ZnSO_4_·H_2_O), 120 mg; I (CaI_2_O_6_), 0.7 mg; and Se (Na_2_SeO_3_), 0.48 mg. ^3^ SID, Standardized ileal digestibility. ^4^ STTD, Standardized total tract digestible.

**Table 2 animals-11-02186-t002:** Primers used for quantitative real-time PCR ^1^.

Primer	Sequence (5′–3′)
β-actin-F	CCTGAACCTCTCATTGCCA
β-actin-R	AGGGCCGTGATCTCCTTCTG
IL1-β-F	GAIAGTGCTTCGTGCTGGAGT
IL1-β-R	ACTGGCATCTGCCCAGTTC
IL-8-F	ATGAGTCTTAGAGGTCTGGGT
IL-8-R	ACAGTGAGGGCTAGGAGGG
IL-10-F	GCATCCACTTCCCAACCA
IL-10-R	GCAACAAGTCGCCCATCT
TNF-α-F	GAAGCAGCGTTTGGGAGTG
TNF-α-R	GTTGTGGGACAGGGTAGGG
GM-SCF-F	GCAATTTCACCAAACTCAAGG
GM-SCF-R	CTCATTACGCAGGCACAAAAG
TGF-β-F	TTGGGACTTGTGCTCTAT
TGF-β-R	AGTTCTGCTGGGATGTTT

^1^ IL-6, interleukin-6; IL-8, interleukin-8; IL-1β, interleukin-1β; IL-10, interleukin-10; GM-CSF, granulocyte-macrophage colony stimulating factor; TGF-β, transforming growth factor-β; TNF-α, tumor necrosis factor-α.

**Table 3 animals-11-02186-t003:** Effect of niacin on growth performance in weaned piglets.

Item	CON	NA	MPN	*p*-Value
ADG (g/d)	101.13 ± 9.41 ^ab^	116.80 ± 9.12 ^a^	83.26 ± 6.26 ^b^	0.042
ADFI (g/d)	192.34 ± 7.18	203.79 ± 7.90	189.96 ± 4.28	0.328
G/F	0.52 ± 0.04	0.57 ± 0.03	0.46 ± 0.02	0.055
Diarrhea incidence (%)	29.68 ± 10.05	24.74 ± 9.42	31.04± 11.09	0.468

Values are expressed as the mean ± SD, *n* = 8/treatment. ^a, b^ Values with different superscripts in the same row differ significantly (*p* < 0.05). The amount of feed consumed per pen was recorded every day to determine the average daily feed intake (ADFI) on an 88% dry matter basis. All of the piglets were weighed individually at the beginning and the end of the experiment to calculate the average daily gain (ADG). The gain to feed ratio (G/F) was calculated based on the weight gain and feed intake (based on 88% dry matter) during the whole experiment period. The diarrhea rate was calculated as the number of piglets with diarrhea/(total numbers of piglets × days) × 100. The fecal score was recorded, as described by Marquardt et al., with a score from 0 to 3 (0: normal feces, 1: moist feces, 2: mild diarrhea, and 3: severe diarrhea). The pigs with a fecal score of 2 or higher were considered to suffer diarrhea [30]. ADG, average daily gain; ADFI, average daily feed intake; G/F, gain to feed ratio; CON, control group; NA, niacin group; MPN, GPR109A antagonist group.

**Table 4 animals-11-02186-t004:** The niacin and nicotinamide content in the intestinal digesta and serum.

Item	CON	NA	MPN	*p*-Value
Niacin content				
Jejunum, ng/mL	65.84 ± 19.01 ^ab^	99.70 ± 15.48 ^a^	38.93 ± 4.39 ^b^	0.03
Ileum, ng/mL	103.04 ± 18.84	114.07 ± 23.94	74.39 ± 18.97	0.44
Colon, ng/mL	821.95 ± 68.20	963.29 ± 164.83	618.74 ± 76.06	0.09
Nicotinamide content				
Serum, ng/mL	29.54 ± 9.29 ^b^	41.38 ± 15.90 ^a^	18.36 ± 6.17 ^b^	<0.05

Values are expressed as the mean ± SD, *n* = 8/treatment. ^a, b^ Values with different superscripts in the same row differ significantly (*p* < 0.05). CON, control group; NA, niacin group; MPN, GPR109A antagonist group.

## Data Availability

The data presented in this study are available from the corresponding author upon reasonable request.

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
