# Peer review of "Effect of Niacin on Growth Performance, Intestinal Morphology, Mucosal Immunity and Microbiota Composition in Weaned Piglets"

_animals, 2021, doi:10.3390/ani11082186_

Round 1

Reviewer 1 Report

With regard to my previous report, the authors have made significant changes to the manuscript that do improve it. However, can the authors explain why the melting curves, now put in supplementary files, have always two peaks per gene? 

Reviewer 2 Report

The manuscript has been deeply revised. The authors added important information in M&M section and in the results section increasing the manuscript “value”. The Research study have been well done and describe; I report some minor revisions below:

Line 31: ...allocated into 3 treatment groups...

Line 66: please define the pharmacological level (such as ppm or ppb)

Line 112:... to... into....

Line 124-127: Please modulate the sentence making it more clear, such as: ... pigs were allotted in cages contained 4 piglets with slatted floors...

Line 127: in the sentence before the authors reported that piglets were fed 4 times.

Line 133: "medium BW" I suggest to modify this sentence- better specify average BW indicating the kg and SD.

Line 138: body weight (BW) please check in the whole text the acronyms which should be specify the first time is used in the text.

Table 2: I suggest to use "niacin additive" instead of "niacin in ingredient"- Basal diet ingredient instead and niacin content instead

Line 183: please check capital letters.

Line 191: please check the correct nomenclature and acronyms of genes in GeneBank- NCBI

Line 263- 268: I suggest to author to move this section after zootechnical performance, closer to the other intestinal results.

Table 4: how authors recorded Diarrhea incidence? did they use a score ore presence or absence of diarrhea? please add this information in the material and methods.

line 299- 301: figure 2 do not report significancy only for TGF-B, please add letters also for the other genes.

Line 303: check the uppercase.

Figure 2: add letters for each panel

Line 323: Figure 2- in the text is reported Figure 2B- please change numbers or add B

I suggest to choose one way to present data about gene expression, Both fold change and relative expression are correct but in my opinion report both seems redundant.

Line 408: better use verbs in past tens

Line 4'9-410: The sentence is not clear, I suggest to modify the sentence

Line 420: Please, define VH:CD

Line 426-427: Please modulate the sentence making it more clear such as: " where with high.."

Line 426-430: Please add citations.

451-454: I suggest to add literature related pig. I can understand that niacin has benne largely studied in ruminants but I am confident that there are publication that have studied niacin in pigs intestine.

Reviewer 3 Report

Most of my comments were addressed appropriately. After giving it some thaught I came to the conclusion to recommend acceptance of the manuscript after further minor revision, although the experimental approach has a serious flaw, which is the fact that no group was introduced that supplies high niacin in the presence of the inhibitor. That is generally not a good practice. Finally, what has to happen during you minor revision is that you analyze your feed, for dry matter, crude nutrients and niacin and to provide info on the analytical procedures in your M&M section. It is inappropriate to perform a feeding trial and to not provide feed quality based on chemical analysis gut feed table estimation.

I think the study might provide some insight that has the potential to direct future research on the matter. Below is my response to some of your answers. I followed the numbering from your response letter.

Points 1 and 6: I already mentioned my concerns on this matter above. You may want to consider this in your future research endeavors. Was the niacin in your feed based on analysis or estimation from feed tables? Your target nutrient should always be addressed by analysis rather than estimation.

Point 19: That is a bit problematic I find, since this is a feeding trial and obtaining chemical information on your feed is the most basic thing to do. In generally, feed dry matter and crude nutrients should be analyzed together with the nutrient under study, in your case niacin. This is crucial to demonstrate that isoenergetic and isonitrogenous conditions have been met. This must be provided from my point of view. Also, highlight the chemical procedures used in your M&M section.

Point 23: That is of course valuable information and should be provided. But what about endogenous markers of niacin status? That was what I meant in the first place.

Point 28: I think the reason why you missed the significance here is the statistical approach used. Fecal scores are no parametrical data and presumably not normally distributed. I would recommend to analyze the data again applying a non-parametrical procedure.

Point 29: Thank you for this clarification. You may want to include this into your discussion.

Point 30: This should be made very clear in the definition of your study goals at the end of your introduction.

Author Response

This manuscript is a resubmission of an earlier submission. The following is a list of the peer review reports and author responses from that submission.

Round 1

Reviewer 1 Report

Liu et al. describe the effect of niacin on performance and gut health of piglets around the critical time of weaning. I think the paper is interesting for the field, but I still have some remarks I like to see addressed.

  • The authors compare a CON diet, a basal diet containing 9.6mg/kg niacin, a NA group, containing in total 30mg/kg and a MPN group, in which they use a niacin receptor antagonist. However, they describe the antagonist supplemented at mg/kg body weight/d where they don’t express the feed of the other two diets (CON, NA) in mg/kg body weight/d. Can the authors explain for a better understanding? Is the feed given restricted? Are the authors using individual feed measurements per piglet? This information lacks in Material and Methods.
  • Is gender considered when dividing piglets in groups? The 8 replicates, are they replicates over time?
  • In the description of Table 1 formulation, I would use the exact 9.6mg/kg in the description of or the Premix, as to make it consistent.
  • Where melting curves used to check the product in the qRT-PCR. If so, please mention.
  • Why did the authors use only one internal control gene, and why was beta actin chosen?
  • In the statistical analysis, change Turkey tot Tukey
  • In result section 3.3 change mRAN abundance to mRNA abundance
  • Why did the authors chose to show protein data in a table with a comparison between NA and CON and between MPN and NA? Why not showing the relative protein abundance in a similar way as the mRNA abundance in figure 2? I would find that easier to compare.
  • In the discussion, the authors state that it is likely that the niacin amount chosen, or the duration of the treatment might have been to low, or to short. How did the authors chose the amount and time for the experiment?

Reviewer 2 Report

General comment

The manuscript aimed to investigate the effect of niacin and antagonist for the niacin receptor GPR109A on growth performance and intestinal health on weaned piglets.

The study is interesting because niacin could modulate inflammatory response and regulate and improve intestinal health.   

The manuscript needs deep revisions, English should be revised by native speaker, the manuscript is lacking essential information regarding experimental protocols and also the statistical analysis needs more explanation. More comments are reported below:

Title: why authors used “weaning piglets” instead of use weaned? In Material and methods, they reported that animals were weaned.

Abstract

Line 27: I suggest avoiding this sentence. Then, authors did not cite this article in the text- only on line 122 they reported (26) as citation.

Line 30: Please, add SD.

Line 36: did authors found differences among treatments and control?

Line 41:

Intestinal mucosa: please specify the intestinal tract.

In comparison...whit other experimental groups? Please, specify.

Line 44: Please specify which treatment groups.

Line 49: No results about growth performance has been showed or described in the abstract. Please add this information even if was reported in a previous publication.

Introduction

Line 64: "accumulation evidence", I suggest to change this sentence to ...Literature data...or... Literature studies...

Line 68: Kwon et al. (12). See authors guideline.

Line 71: please define pro and anti-inflammatory cytokines- acronyms.

M&M

Materials and methods are lacking of essential information regarding the experimental trial, such as:

- I do not see any previous research study cited in which is explained animals' rearing condition.

- Where was conducted the experiment? research center? experimental farm? commercial farm?

- Where were allotted piglet? cages? type of floors?

- When was distributed the experimental diets? water? free access? Ad libitum?

- How and when authors measured ADG and AFI?

- How authors euthanized animals? Procedures? this is a very important aspect- how many animals? all the animals enrolled? When? Where?

Table 1: Define STTD

Line 104: The age of piglets is missing.

Line 120: Authors did not specify which segments of intestine were collected.

Table3: The sampling time is missing, how and when growth performances were recorded? one time- several times?

Line 178: Turkey’s test? maybe tukey test?

Results

Table3: The sampling time is missing, how and when growth performances were recorded? one time- several times?

Table 4: Why authors decide to present only NA vs CON and MPN vs NA? MPN and CON is missing.

I highly suggest to present all the combinations, or better to have three columns with CON, NA and MNE an relative p value using in the statistical model all three groups, otherwise justify the choice. 

Colonic microbiota section: the figure is not well visualized, is very difficult to red all boxes reported (Fig. 3-4-5) I suggest to make the figures more clear.

Discussion

The results are logically disused in the discussion section, this section does not need deep revisions.

Conclusion

  …. GPR109A-dependent pathway, ultimately maintains intestinal health and relieves weaning stress….

I suggest to modulate this sentence, better explain “GPR109A-dependent pathway”- Why?

I am concern about using “relieves weaning stress”, authors did not measure any marker of stress even can of diarrhea  (which can be a sign of stress) or blood marker of stress (such as cortisol).

Reviewer 3 Report

Abstract

  • Highlight the niacin concentration in the MPN diet. This also refers to your material and methods section. As for now, it reads as the MPN was supplemented on a low-niacin diet, which would be problematic for obvious reasons.
  • Demonstrate the relative results obtained in your treatment group relative to both control groups, not only negative control or MPN, respectively.

Introduction

  • Lines 63-77. That is a rather descriptive display of niacin effects. Overall, the way it is currently written suggests that niacin by itself promotes these effects, although it is very likely that some effects may be of secondary nature. After all, niacin is an essential nutrient; hence, deficiency is associated with metabolic malfunction. It would be highly appreciated if the authors could share more details on the precise mode-of-action. For example, how exactly does niacin supplementation attenuate lung inflammation and survival rate during sepsis and were these results obtained relative to a niacin-deficient control group or did they suggest some sort of pharmacological effect? This should be addressed by highlighting specifically, which metabolic functions are niacin dependent and how does that relate to the positive effects observed in earlier studies.

Material and Methods

  • No information is given with respect to the age of animals at experimental start as well as the sex of animals.
  • Which effects size and statistical power did you expect from the experimental setup with regard to the different kinds of data obtained? This should also be highlighted in the text.
  • State specifically, the experimental units for the different datasets. Some data may be analyzed pen-wise whereas others based on individual animals. Hence, the effective sample size is different for different measures and must be specifically stated. This issue is also connected to my former comment.
  • It has been mentioned earlier that the MPN diet’s niacin concentration should be highlighted. Overall, it could be argued that the design would have needed four instead of three groups, which the MPN being administered in the presence of low and high niacin, thereby allowing a more precise dissection of effects compared to the NC and niacin groups.
  • Niacin concentration should be highlighted in your feed table.
  • Highlight also the analyzed dry matter concentration in your feed table.
  • Please do not use kcal as energy unit and switch to SI units, in this case kJ.
  • Using a 1% agarose gel for the separation of RNA is rather low, considering the size of the nucleic acids. Isn’t a higher percentage more straightforward?
  • Please follow the MIQE guidelines for the communication of applied qPCR methods. Also, indicate why you just used one reference gene, which is against the advised practice of geometric averaging of multiple reference genes. Please also defend why beta-actin was your choice, keeping in mind that it is not sufficient to just state earlier data that have used it. Reference genes must be evaluated statistically for every new qPCR experiment.
  • Please show the average yield and purity obtained for RNA samples of different treatment groups. Also provide (in the supplementary material) and exemplary gel picture demonstrating the RNA integrity of RNA from different feeding groups.
  • Barcode sequencing. The information provided appears a bit superficial. Especially the bioinformatics behind the data should be highlighted in more detail, indicating used algorithms, software packages etc.. Please keep in mind, that all information must be accessible to the reader, to allow a direct reproduction of your findings. So write your methods in a way that allows for that. Just think about what you would miss as an information if you tried to rebuild another groups approach. Of course, if other sources provide this information you could cite them instead, but only if the information given there is thorough enough. This comment refers to your M&M section in general.
  • Statistical analysis. “Replicates (n = 8) served as the experimental units.”. Does this really refer to all datasets, including those based on tissue samples from individual animals? I would have suspected that e.g. qPCR or barcode sequencing were done on animal-individual sample material, hence, in these cases your experimental unit would have been the individual animal and your sample size would presumably be different from 8/group?
  • Statistical analysis. You wrote that data was presented as mean +/- SEM. First, it is not recommended to use the standard error for the display of in-group variation when the group size is <1000 due to its low susceptibility to outliers and extreme values. Hence, in your case the standard deviation is a better (“more honest”) choice. Secondly, please do not abbreviate the regular standard error as SEM. SEM is the abbreviation for “standard error of means”, which represents the pooled standard error of a (generalized) linear model and is calculated from the mean square error corrected for the respective degrees of freedom. What you meant is SE as an abbreviation of a regular standard error.
  • Statistical analysis. I assume you meant “Tukey’s Test” when mentioning that you used “Turkey’s Test” for mean value comparison.
  • Statistical analysis. “The relative abundance of colonic bacteria and protein expression were assessed using a t-test.” à This statement puzzles me. These measures where obtained in animals from all three groups, right? However, a regular t-test is only able to compare two groups at once. Please explain!
  • Statistical analysis. On which assumptions did you base your definition of fold-change thresholds for protein expression (>1.2, <0.83)? Was a power analysis involved in the decision-making process? In addition, please explain what a fold-change <0.83 is supposed to indicate?
  • No information on feed analysis procedures was provided.
  • Statistical analysis. In your results section, you state that correlation between zootechnical performance and microbial abundance was assessed by using Spearman correlation. Here you state it was Pearson correlation. What is true?

Results

  • Figures and tables must be self-explanatory without knowledge of the text. Hence, all information necessary for their interpretation must be presented, like definition of abbreviations, which statistics were applied, statistical thresholds etc.. Finally, the titles must be specifically highlighting what is presented in a "The effects of X on Y in Z" manner.
  • Please describe the results in the text by considering all groups. For example, duodenal architecture was improved in NA animals compared to both, CON and MPN not only NC.
  • It is odd that no data on the niacin status of animals was obtained although niacin was the nutrient under study.

Discussion

  • Line 282-290. This paragraph pretty much sums up the biggest problem of the present study. Your refer to the problems with weaning stress but there is no indication that your animals were stressed, although with 21d weaning occurred early. Most puzzling for me is the fact that diarrhea scores were not assessed, which would have been interesting since the biggest problem associated with early weaning is in fact the post-weaning diarrhea (“leaky gut syndrome”). Can such data be provided?
  • Lines 291-203. If extra niacin “promotes” zootechnical performance, most likely this indicates that the control group is niacin deficient. Please keep in mind that niacin is an essential nutrient. Although pharmacological effects in response to drastic over-supplementation cannot be ruled out, I would be careful to interpret niacin findings from the perspective of a growth-promoting supplement rather than a nutrient.
  • Lines 304-315. Here and elsewhere, please discuss your MPN findings also with respect to the niacin supply level administered in the respective diet. With regard to this, it reduces the scientific merit of this study that only one of the niacin levels (that of CON) was used in the presences of MPN but not in the presence of the higher niacin dose. This drastically impairs the resolution of this dataset.
  • Lines 316-337. This comment also refers to that on your introduction. Given the fact that niacin is a nutrient under study for decades now, I find it unsatisfying that there is only highlighted that supplementation of niacin caused an up- or down-regulation, respectively, of certain parameters. Please extend your discussion on the matter, by introducing information on the specific mode-of-action. Given the role of niacin in biology, how, for example, does it regulate the expression of cytokines? Is it per-se a signaling molecule that activates protein biosynthesis pathways? Please try to address this issue more thoroughly.
  • The data should also be discussed in light of the phenotype of your animals. You refer a lot to the potential health benefits of niacin with respect to the gut. However, was there any indication that gut health was impaired in the pigs?
  • Lines 338-360. Same as for the previous comment. Is there any indication of how exactly niacin affects the microbial composition? Is that a direct effect on microbes (which would be a pharmacological effect) or does the niacin status of the animal affect the mucosal composition and therefore the host-microbe interaction?

Conclusions

  • It is yet not clear to me what precisely makes the present study unique compared to the earlier studies on the matter? It appears that for most of your findings your find also a reference from the literature that confirms you findings. Hence, what makes this study novel? This question must be addressed before this manuscript can be reconsidered for publication.
